# Biomarkers of Micronutrients in Regular Follow-Up for Tyrosinemia Type 1 and Phenylketonuria Patients

**DOI:** 10.3390/nu11092011

**Published:** 2019-08-27

**Authors:** Kimber van Vliet, Iris L. Rodenburg, Willem G. van Ginkel, Charlotte M.A. Lubout, Bruce H.R. Wolffenbuttel, Melanie M. van der Klauw, M. Rebecca Heiner-Fokkema, Francjan J. van Spronsen

**Affiliations:** 1Division of Metabolic Diseases, Groningen, Beatrix Children’s Hospital, University Medical Center Groningen, University of Groningen, 9700 RB Groningen, The Netherlands; 2Department of Endocrinology, Groningen, University Medical Center Groningen, University of Groningen, 9700 RB Groningen, The Netherlands; 3Laboratory of Metabolic Diseases, Groningen, Department of Laboratory Medicine, University Medical Center Groningen, University of Groningen, 9700 RB Groningen, The Netherlands

**Keywords:** Phenylketonuria, Tyrosinemia type 1, nutritional status, micronutrients, vitamin D, vitamin B12

## Abstract

Phenylketonuria (PKU) is treated with dietary restrictions and sometimes tetrahydrobiopterin (BH4). PKU patients are at risk for developing micronutrient deficiencies, such as vitamin B12 and folic acid, likely due to their diet. Tyrosinemia type 1 (TT1) is similar to PKU in both pathogenesis and treatment. TT1 patients follow a similar diet, but nutritional deficiencies have not been investigated yet. In this retrospective study, biomarkers of micronutrients in TT1 and PKU patients were investigated and outcomes were correlated to dietary intake and anthropometric measurements from regular follow-up measurements from patients attending the outpatient clinic. Data was analyzed using Kruskal–Wallis, Fisher’s exact and Spearman correlation tests. Furthermore, descriptive data were used. Overall, similar results for TT1 and PKU patients (with and without BH4) were observed. In all groups high vitamin B12 concentrations were seen rather than B12 deficiencies. Furthermore, all groups showed biochemical evidence of vitamin D deficiency. This study shows that micronutrients in TT1 and PKU patients are similar and often within the normal ranges and that vitamin D concentrations could be optimized.

## 1. Introduction

Phenylketonuria (PKU, OMIM 261600) and tyrosinemia type 1 (TT1, OMIM 276700) are both inborn errors affecting the phenylalanine-tyrosine degradation pathway necessitating a comparable dietary treatment. PKU affects approximately 1:18.000 newborns and is caused by a defect in the enzyme phenylalanine hydroxylase. Biochemically, this leads to high phenylalanine and low/normal tyrosine concentrations. Untreated, high phenylalanine concentrations will lead to severe mental impairments [1]. Treatment nowadays consists of dietary restriction of phenylalanine, amino acid supplements, and, if patients are responsive, the drug sapropterin dihydrochloride, which is the synthetic form of tetrahydrobiopterin (BH4) [2]. BH4 acts as a pharmacological chaperone of phenylalanine hydroxylase, improving the conversion of phenylalanine into tyrosine, thereby lowering the phenylalanine concentrations [3]. Various studies have been conducted on the effect of this dietary treatment and of BH4 on the nutritional and biochemical parameters in PKU, reporting that PKU patients are at risk for micronutrient deficiencies such as vitamin B12, folic acid, selenium, zinc and iron [4,5,6]. This also applies to PKU patients who are treated with BH4, in whom deficiencies may result from their liberalized diet [7,8,9].

Tyrosinemia type 1 is caused by a deficiency of the enzyme fumarylacetoacetate hydrolase and is much rarer than PKU, affecting approximately 1:100.000 newborns. The deficient enzyme leads to the accumulation of toxic metabolites, which in turn cause renal tubular dysfunction, neurological porphyria-like crises, liver failure and liver cancer [10]. In the past, the only definitive form of treatment was a liver transplantation, usually at a very young age, in order to survive. Since 1992, however, a new treatment option has become available called 2-(2-nitro-4-trifluoromethylbenyol)-1,3-cyclohexanedione (NTBC), which blocks the activity of 4-hydroxyphenylpyruvate dioxygenase, an enzyme upstream of the metabolic defect [11]. Treatment nowadays consists of NTBC paired with a dietary restriction of tyrosine and its precursor phenylalanine. On this treatment, outcome has improved tremendously. Since the NTBC-dietary treatment, the focus of research has shifted towards neurocognitive deficits that are observed and their possible cause [12,13,14,15,16]. At present, however, little research has been conducted on the nutritional and biochemical parameters in TT1. Although PKU and TT1 are different diseases, they are treated with a comparable protein-restricted diet and amino acid supplements. This raised the question whether TT1 patients may also be at risk for micronutrient deficiencies as in PKU patients.

Therefore, the aim of this retrospective study was (1) to compare outcomes of regular biochemical follow-up measurements of TT1 and PKU patients (both with and without BH4 treatment); (2) to investigate possible deficiencies or excesses in blood of TT1 patients, PKU patients treated with BH4 (PKU-BH4), and PKU patients not treated with BH4 (PKU-nBH4); and (3) to assess possible correlations between dietary intake and metabolic control and these regular follow-up outcomes.

## 2. Materials and Methods

### 2.1. Participants

In total, 12 TT1 patients (mean age 13.5 ± 9.9, 75% male) and 92 PKU patients (mean age 24.5 ± 13.9, 45% male) were included. All patients were treated at the University Medical Center Groningen, The Netherlands. Of the PKU patients, 33 patients (36%) received BH4 at the time of assessment. BH4 dosages ranged from 100 mg to 1400 mg a day. The TT1 patients were all treated with NTBC and dietary restriction of phenylalanine and tyrosine. NTBC dosages ranged from 0.51 mg/kg/day to 1.17 mg/kg/day. Both the TT1 and PKU patient groups were treated with several different amino acid mixtures (see Appendix A for information regarding micronutrient and mineral content of amino acid supplements used by our patients). The need for formal ethical review and informed consent was waived by the local medical ethical committee.

### 2.2. Study Parameters

Data were collected retrospectively from patient files. From each patient the most recent blood measurements at the outpatient clinic for regular follow-up were included. All data came from blood samples collected between 14 August 2017 and 29 April 2019. Patients who had no blood samples taken at the hospital in that period were excluded from analyses. Concentrations were considered abnormal if they were found to be outside the reference range of the hospital laboratory information system.

All blood measurements during regular outpatient clinic visits were collected, which encompassed: leucocytes, hemoglobin (Hb), hematocrit (Ht), mean corpuscular volume (MCV), thrombocytes, plasma sodium, potassium, chloride, creatinine, urea, calcium, phosphate, magnesium, albumin, total protein, alkaline phosphatase, ferritin, vitamin B12, methylmalonic acid (MMA), total homocysteine, 25-OH-vitamin D3, pre albumin, and thyroid stimulating hormone (TSH). Furthermore, plasma phenylalanine and tyrosine concentrations were collected. Plasma folic acid, alanine aminotransferase and aspartate aminotransferase concentrations were excluded from analyses, as folic acid concentrations were in the majority of cases derived from hemolytic blood samples, which can give falsely elevated results, and alanine aminotransferase and aspartate aminotransferase were only occasionally collected. For the remaining blood parameters, in total, 17% of our blood value data were missing. From the day of blood collection, the standard deviation scores (sds) of anthropometric measurements (height sds, weight sds, and body mass index (BMI) sds) were collected, as well as dietary information, which included total protein intake, natural protein intake, protein intake from amino acid supplements and total protein intake per kg body weight.

### 2.3. Statistical Analyses

Differences in anthropometric measurements and dietary intake between the groups were investigated using Kruskal-Wallis tests, since data were not normally distributed. The Mann–Whitney U test was used as a post-hoc test, and its *p*-value was corrected for multiple comparisons according to Bonferroni. Since some reference values from the blood measurements differed between age and gender, categorical variables were created (below normal, normal, and above normal). Because of the low expected cell frequencies for these measurements, Fisher’s exact tests were performed to determine differences between TT1, PKU-nBH4 and PKU-BH4 patients and 2-sided *p*-values were adhered. Additionally, percentages of these abnormal values were calculated.

To investigate possible correlations between outcomes and dietary intake, Spearman correlation tests were performed. Since dietary intake differs between age groups, the percentage of natural protein intake from total protein intake was calculated and used for these analyses. Lastly, Spearman tests were used for investigating possible correlations between metabolic control (phenylalanine and tyrosine concentrations) and observed deficiencies and/or excesses. All statistical analyses were performed using IBM SPSS Statistics 23. *p*-values < 0.05 were considered statistically significant.

## 3. Results

### 3.1. Tyrosinemia Type 1 (TT1) versus Phenylketonuria (PKU)-nBH4 versus PKU-BH4

Descriptives are summarized in Table 1. No significant differences were observed between the groups regarding the anthropometric measurements. When comparing percentage of natural protein intake between the three patient groups, Kruskal-Wallis tests showed a significant difference (*p* < 0.001). Pairwise comparisons then showed that differences existed between PKU-BH4 and PKU-nBH4 patients (*p* < 0.001) and between TT1 and PKU-BH4 patients (*p* = 0.033), with higher natural protein intake for PKU-BH4 patients, as expected. After Bonferroni correction for multiple comparisons only the difference between PKU-BH4 and PKU-nBH4 remained statistically significant (*p* < 0.001).

To investigate differences in blood measurements between the patient groups (TT1, PKU-nBH4, or PKU-nBH4) Fisher’s exact tests were performed using the categorical data. A significant difference between the groups was observed regarding MCV levels (*p* = 0.038), suggesting more often higher MCV levels in the PKU-BH4 group compared to the TT1 and PKU-nBH4 groups. Other than this, no significant differences between the three groups were observed.

### 3.2. Deficiencies or Excesses

Table 2 shows percentages of patients with values below or above the reference ranges in the different patient groups (TT1, PKU-BH4, and PKU-nBH4). High vitamin B12 concentrations were often observed (Figure 1). All patient groups appeared to be prone to having an excess of vitamin B12, with respectively 36%, 33% and 20% of the patients having a concentration above the reference limit. Elevated levels of MMA and total homocysteine were occasionally observed (in 5 and 2 patients respectively). Elevation was always in only one of these markers, never in both.

Next to this, low vitamin D concentrations were present in our sample (Figure 1). This was the case for all patient groups, and especially for both groups of PKU patients (8%, 29%, and 24% respectively). Since vitamin D levels are known to differ depending on time of the year [17], Figure 2 shows the vitamin D concentrations observed per month.

### 3.3. Correlations with Dietary Intake

When correlating the blood concentrations to the percentage of natural protein intake, correlations were observed for vitamin B12 (r = −0.277, *p* = 0.013), total homocysteine (r = 0.346, *p* = 0.002), and vitamin D (r = −0.232, *p* = 0.026). These correlations suggest that vitamin B12 and D concentrations are lower in patients with a higher percentage of natural protein intake, and that total homocysteine concentrations are higher with more natural protein intake. No other correlations were observed between protein intake and blood concentrations.

### 3.4. Correlations with Phenylalanine and Tyrosine Concentrations

Spearman correlation tests were performed to investigate possible correlations between the vitamin B12 and vitamin D concentrations and phenylalanine and tyrosine concentrations. Separate analyses were performed for TT1 and PKU patients because of the opposite biochemical profile. For TT1 a significant correlation was observed between tyrosine and vitamin B12 (r = −0.627, *p* = 0.039) and a trend was observed for tyrosine and vitamin D (r = −0.566, *p* = 0.055). For PKU patients significant positive correlations were observed between tyrosine concentrations and both vitamin B12 (r = 0.279, *p* = 0.013) and vitamin D (r = 0.319, *p* = 0.002). Phenylalanine concentrations showed no significant correlations in neither the TT1 or the PKU group.

When investigating PKU-nBH4 and PKU-BH4 patients separately, PKU-BH4 patients showed no significant correlations. PKU-nBH4 patients showed a significant correlation between tyrosine and vitamin D (r = 0.378, *p* = 0.003). Furthermore, trends were observed between tyrosine concentrations and vitamin B12 (r = 0.272, *p* = 0.061) and between phenylalanine and vitamin D (r = −0.255, *p* = 0.054).

## 4. Discussion

This study describes micronutrients collected for regular follow-up in TT1 and PKU patients in our hospital. The biochemical and micronutrient status of TT1 patients was largely comparable to that of PKU patients, as might be expected for the dietary treatment of PKU-nBH4 and TT1 is comparable. Especially in PKU patients, and to a lesser extent in TT1 patients, vitamin B12 concentrations were elevated, while vitamin D generally appeared to be lower than normal. In both diseases, both vitamin concentrations could to some extent be related to the natural protein intake and the tyrosine rather than the phenylalanine concentrations.

Before discussing our findings in more detail, some methodological issues need to be addressed. The group of TT1 patients in this study was relatively small compared to the group of PKU patients, which is due to the difference in incidence. This smaller sample size for the TT1 patients may have been insufficient to generate significant results. Since this is a retrospective study, we could only use data that had been requested by the clinician at the time of the outpatient visit. As a consequence, 17% of the blood values were missing. Dietary information was collected from patient files. However, these may not completely reflect the actual situation since patients may not always count their protein intake or have low treatment compliance [18,19]. It has been shown that PKU patients with high adherence to diet may have significant differences in biochemical markers compared to patients with low adherence to diet [5]. In our study, adherence could not be taken into account, and these possible differences could not be investigated. Furthermore, differences in blood concentrations for different amino acid mixtures could not be investigated because of the low number of patients per product. Lastly, some possibly relevant markers, such as blood concentrations of zinc and selenium [20], were not assessed, since these measurements are generally not included in regular follow-up in our center.

With regard to the findings, several studies have shown that PKU patients, both with and without BH4 may be at risk for developing vitamin B12 deficiency especially when adherence to diet is suboptimal [4,6,9]. For TT1 patients, evidence on possible B12 deficiencies is currently lacking. Remarkably, in our sample, vitamin B12 concentrations in TT1 and PKU patients were elevated rather than lowered. This could possibly be caused by a high dietary adherence and thus adequate intake, as also observed previously [5,20]. It should be noted, however, that MMA and total homocysteine are better markers of vitamin B12 status than B12 concentrations themselves, as a vitamin B12 deficiency can exist even when B12 concentrations are normal or high [4,21]. Since increased MMA and/or total homocysteine concentrations, suggesting a B12 deficiency, were found in a small number of patients (all having normal/high B12 concentrations), this may point toward a suboptimal B12 status in these patients. Vugteveen et al. suggested that vitamin B12 intake by amino acid mixture may not result in optimal absorption of vitamin B12 [21], which may in part explain our observed results.

Furthermore, our study suggests that a biochemical vitamin D deficiency may arise in approximately 25%–30% of the PKU patients, both PKU-nBH4 and PKU-BH4 patients (29% and 24%, respectively). A recent study by Kose and Arslan showed an even larger percentage of vitamin D deficiency in PKU patients in Turkey (53.6%) [5]. However, they also showed a similar percentage of healthy controls (47.2%) with this deficiency. In addition to this, a Dutch study investigating the prevalence of biochemical vitamin D deficiency in the normal population reported deficiencies in almost 60% of healthy individuals in the winter and roughly 30% in the summer [17]. These numbers considered, our PKU patients are doing relatively well. However, reference values in our hospital adhere to a lower limit of 50 nmol/L, while it has been suggested that, optimally, values should be above 75 nmol/L [17]. The use of this favorable reference range would lead to the finding that vitamin D levels were suboptimal in 62.5% of our patients (including 7/12 TT1 patients), even in some patients who were reported to use vitamin D supplements. PKU patients have also been shown to suffer from bone problems, such as reduced mineral bone density [4,22]. Although the pathogenesis of this has not been fully elucidated yet, we recommend, as a start, to optimize vitamin D concentrations in these patients. TT1 patients on the other hand, are known to develop vitamin D-resistant rickets, if untreated [23]. For TT1 patients the effect of vitamin D supplementation might, therefore, be different than the effect on PKU patients, but this needs further investigation. Both vitamin D and B12 concentrations were correlated to percentage of natural protein intake. The results suggest that higher intake of proteins from supplements result in higher vitamin concentrations. This may be due to the high vitamin content of amino acid supplements as also suggested previously [24]. Extra attention could, therefore, be paid to patients on a higher natural protein diet or patients who discontinued their amino acid supplements.

With tyrosine being the precursor of thyroxin, it can be hypothesized that altered tyrosine concentrations, as observed in PKU and TT1, may have an influence on thyroxin availability which can be assessed by measuring TSH levels. Our results did not show clear aberrations in TSH levels, which is in contrast with Sumanszki et al. who did observe lower TSH levels in adult PKU patients compared to controls [25]. Therefore, the possible effect of tyrosine (and phenylalanine) concentrations on thyroid function should be investigated in more detail.

Lastly, tyrosine concentrations appeared to be correlated to the vitamin B12 and vitamin D concentrations. For TT1, higher tyrosine levels (suggestive of less treatment adherence) indicated lower vitamin concentrations. Similarly, in PKU patients, higher tyrosine concentrations (suggestive of better treatment adherence) correlated with higher vitamin B12 and vitamin D. These findings underline the conclusion of Vugteveen et al. that the vitamin B12 concentrations may be related to the intake of amino acid supplements [21]. Literature describes negative correlations between vitamin B12 levels in PKU and phenylalanine levels, however, correlations with tyrosine have not been investigated previously [5,6]. Thus, although correlations with phenylalanine levels were not observed in our sample, both our results and the results from literature suggest a positive effect of treatment adherence on micronutrient status.

## 5. Conclusions

Diet therapy is important in the treatment of both PKU and TT1 patients. For many years, there have been concerns about nutritional deficiencies, however, the results of this study show that this is often not the case. The biomarkers of micronutrients of TT1 patients were largely comparable to those of PKU patients, and generally within the reference ranges. Vitamin B12 concentrations were high rather than low, while vitamin D concentrations were often below the reference range for which attention should be paid to the intake of protein supplements while specific supplementation can also be considered.

## Figures and Tables

**Figure 1 nutrients-11-02011-f001:**
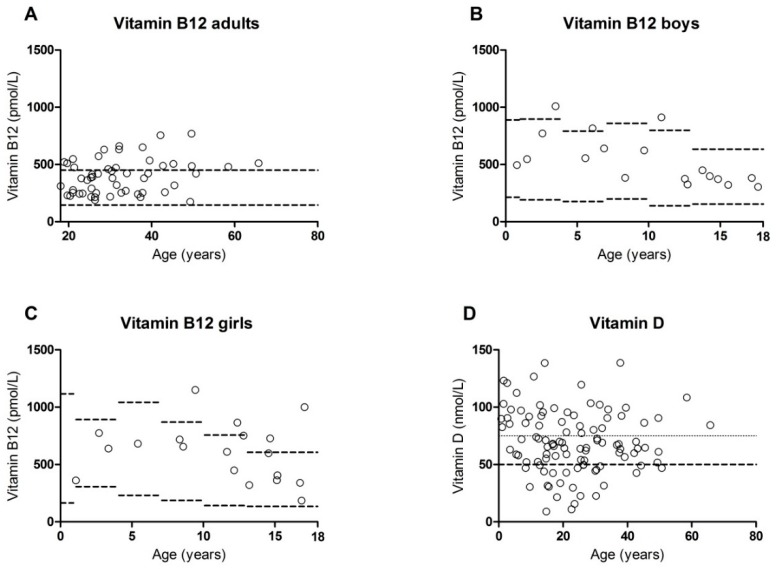
Distribution of vitamin concentrations in patients with reference ranges (depending on age). (**A**): Vitamin B12 concentrations in adult patients. (**B**): Vitamin B12 concentrations in male patients aged <18 years. (**C**): Vitamin B12 concentrations in female patients aged <18 years. (**D**): Vitamin D concentrations in patients with lower reference limit of 50 nmol/L and recommended limit of 75 nmol/L.

**Figure 2 nutrients-11-02011-f002:**
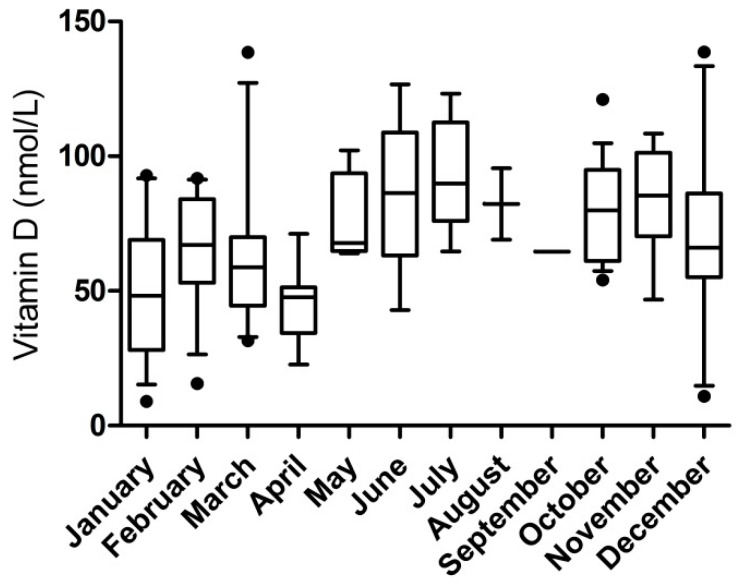
Vitamin D concentrations per month. Whiskers indicating 10–90 percentiles.

**Table 1 nutrients-11-02011-t001:** Descriptive data. For each patient group the minimum and maximum observed values were included, as well as the number of patients for which that value was collected.

Measurement		TT1			PKU-nBH4			PKU-BH4		
	Unit	Min–Max	Median	N	Min–Max	Median	N	Min–Max	Median	N
Age	years	0.79–28.54	11.85	12	1.49–65.77	25.73	59	1.08–43.18	19.15	33
Phenylalanine	µmol/L	18– 61	43	12	63–1757	667	58	84–676	313	33
Tyrosine	µmol/L	237–742	489	12	24–149	60	58	29–107	58	33
Anthropometrics										
Height	sds	−1.03–0.88	-0.18	11	−2.64–2.48	-0.46	50	−2.42–1.00	−0.17	32
Weight	sds	−0.61–1.91	0.70	12	−1.61–4.72	0.97	36	−1.59–3.84	0.25	26
BMI	sds	0.24–2.48	1.20	11	−1.21–4.03	1.08	51	−2.10–3.60	0.61	31
Hematology										
Leucocytes	10^9^/L	4.2–10.3	6.9	12	3.6–13.9	6.8	54	4.4–11.4	6.9	32
Hemoglobin	mmol/L	6.4–10.9	8.3	12	6.7–10.8	8.5	55	6.8–10.5	8.5	33
Hematocrit	L/L	0.31–0.49	0.38	12	0.31–0.49	0.40	52	0.33–0.47	0.41	30
MCV *	fL	74.1–93.8	84.6	12	74.2–95.8	88.1	55	73.8–97.7	89.8	33
Thrombocytes	10^9^/L	114 - 464	281	12	138 - 470	258	54	151–412	266	32
Micronutrients and related parameters										
Sodium	mmol/L	137–142	139	11	135–144	140	54	138–142	140	30
Potassium	mmol/L	3.6–5.0	3.9	11	3.4–4.5	4.0	54	3.6–4.6	4.0	29
Chloride	mmol/L	100–105	104	7	101–107	104	24	102–106	104	12
Creatinine	µmol/L	14–84	44	12	25–87	61	52	32–90	63	29
Urea	mmol/L	2.1–7.2	3.7	10	1.2–6.5	4.0	53	1.8–5.8	4.0	30
Calcium	mmol/L	2.37–2.52	2.43	4	2.28–2.56	2.43	43	2.20–2.61	2.42	24
Phosphate	mmol/L	0.67–1.95	1.37	11	0.56–1.65	0.98	51	0.70–1.63	1.06	28
Magnesium	mmol/L	0.74–1.00	0.79	9	0.76–0.97	0.84	43	0.77–0.98	0.85	22
Albumin	g/L	48–48	48	1	43–53	48	18	44–53	49	13
Total protein	g/L	70–78	73	11	58–84	73	50	66–81	74	29
Alkaline phosphatase	U/L	49–434	214	11	46–370	85	42	50–283	83	21
Ferritin	µg/L	16–188	54	11	15–331	52	47	15–351	66	25
Vitamin B_12_	pmol/L	252–1150	630	11	174–1000	423	49	214–682	377	30
MMA	nmol/L	114.8–304.7	168.7	11	47.4–527.8	158.9	56	94.3–562.9	196.5	32
Total homocysteine	µmol/L	4.1–9.6	5.4	11	3.3–17.7	6.3	44	3.6–21.0	7.6	28
25-OH-Vitamin D3	nmol/L	9.1–138.6	61.0	12	10.9–138.7	67.5	59	22.7–99.5	67.8	33
Pre albumin	g/L	0.15–0.48	0.32	11	0.15–0.51	0.30	54	0.17–0.45	0.30	27
TSH	mU/L	0.58–3.88	3.31	10	0.73–6.00	1.71	49	0.46–7.60	1.76	23

Abbreviations: TT1 = Tyrosinemia type 1; PKU = Phenylketonuria; Sds = standard deviation scores; BMI = body mass index; MCV = mean corpuscular volume; MMA = methylmalonic acid; TSH = thyroid stimulating hormone. * Significant difference between all groups with p < 0.05 (Fisher’s exact test).

**Table 2 nutrients-11-02011-t002:** Percentage of patients within the different patient groups with too low/deficient blood concentrations, and too high/excessive concentrations. Percentages are included when the N > 0.

	Patients with a Deficiency			Patients with an Excess		
	TT1	PKU-nBH4	PKU-BH4	TT1	PKU-nBH4	PKU-BH4
	N	N	N	N	N	N
Blood count						
Leucocytes	1 (8%)	2 (4%)	0	0	4 (8%)	3 (9%)
Hemoglobin	0	2 (4%)	1 (3%)	0	0	0
Hematocrit	0	1 (2%)	0	1 (8%)	3 (6%)	5 (16%)
MCV	0	0	0	0	1 (2%)	5 (15%)
Thrombocytes	1 (8%)	4 (8%)	0	2 (17%)	4 (7%)	2 (6%)
Micronutrients and related parameters						
Sodium	0	0	0	0	0	0
Potassium	0	2 (4%)	0	0	0	0
Chloride	0	0	0	0	0	0
Creatinine	1 (8%)	2 (4%)	1 (3%)	0	0	0
Urea	0	5 (10%)	1 (3%)	0	0	0
Calcium	0	0	0	0	0	1 (4%)
Phosphate	1 (9%)	2 (4%)	1 (3%)	0	0	0
Magnesium	0	0	0	0	0	0
Albumin	0	0	0	0	4 (24%)	3 (23%)
Total protein	0	1 (2%)	0	0	1 (2%)	3 (10%)
Alkaline phosphatase	0	0	0	1 (9%)	2 (5%)	1 (5%)
Ferritin	0	2 (4%)	2 (8%)	0	3 (7%)	0
Vitamin B_12_	0	0	0	4 (36%)	16 (33%)	6 (20%)
MMA	0	4 (7%)	0	0	2 (4%)	3 (9%)
Total homocysteine	0	0	0	0	1 (2%)	1 (3%)
25-OH-Vitamin D3	1 (8%)	17 (29%)	8 (24%)	0	0	0
Pre albumin	1 (9%)	6 (11%)	1 (4%)	3 (27%)	4 (8%)	3 (11%)
TSH	0	0	1 (4%)	0	3 (6%)	1 (4%)

Abbreviations: TT1 = Tyrosinemia type 1; PKU = Phenylketonuria; MCV = mean corpuscular volume; MMA = methylmalonic acid; TSH = thyroid stimulating hormone.

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
