# Peer review of "Biomarkers of Micronutrients in Regular Follow-Up for Tyrosinemia Type 1 and Phenylketonuria Patients"

_nutrients, 2019, doi:10.3390/nu11092011_

Round 1
Reviewer 1 Report
This retrospective study sought to describe biochemical follow-up parameters between Tyrosinemia type 1 (TT1) and Phenylketonuria (PKU) patients treated with or without sapropterin. The authors also explore potential deficiencies and excesses in the above groups as well as possible correlations.
Albeit the nature of study, this work addresses an important issue in the scope of two inborn errors of amino acid metabolism given the fact that nutritional therapy remains the cornerstone of PKU treatment and it is also very important in TT1.
Major points:
· Title:
ü The title of the manuscript “Biomarkers of micronutrients and minerals in regular follow-up for Tyrosinemia type 1 and Phenylketonuria patients” should be revised. Taking into account that micronutrients encompasses vitamins and minerals, “micronutrients and minerals” is a redundant title. The same occurs throughout the study and I highly recommend revision.
· Abstract:
ü In the lines 20 and 21, authors say “In this retrospective study, biomarkers of micronutrients and minerals in TT1 and PKU patients are investigated and compared…”. Where are the p-values in the table 1?;
· Statistical Analyses:
ü Mann Whitney is not a post-hoc. test. Please explain that;
ü Bonferroni is a post-hoc test for variables with normal distribution. Please explain the use of the Bonferroni;
· Results:
ü Please clarify if anthropometrics measurements distinguish between different ages. Authors should use z-scores according to WHO for ages under 19;
ü In tables 2 and 3, a great number of parameters cannot be considered micronutrients but biochemical parameters (i.e. alkaline phosphatase);
· Discussion:
ü Please clarify the use of neurotransmittersin the line 176.
Minor points:
· Abstract:
ü In thelines 20 and 21, authors say “In this retrospective study, biomarkers of micronutrients and minerals in TT1 and PKU patients areinvestigated and compared…”. I also suggest putting the verb to be in the past in order to uniformize with the remaining text;
· Conclusions:
ü I suggest the use of diet therapyinstead of dietary treatment in order to avoid repeating the word treatment.
Reviewer 2 Report
This revision is a retrospective with low number of PKU and Tyr I cases. It is remarkable that in the methodology the BH4 dosis for PKU is very high (up to 1400 mg/k/day?).
The methodology is correct.
By means of the results that there is correlation between nutrition supplements and B12 vitamin, so to the adhesion as the authors establish. The problem of deficit of B12 is a social problem but not related to the disease indeed, as the authors estate. The readers could miss the valoration of the parameters with the anthrophometry (z-score or sds height) for PKU-BH4 (-2.4 - 1.0) with so high BH4 dosis.
The results are not specially very novel for these diseases. However, the methodology and the design are well structured.
Round 2
Reviewer 1 Report
I´ve already revised the cover letter and the revised version of manuscript and I consider that the paper is now ready for publication in Nutrients.